# Flexible, Transparent and Highly Conductive Polymer Film Electrodes for All-Solid-State Transparent Supercapacitor Applications

**DOI:** 10.3390/membranes11100788

**Published:** 2021-10-16

**Authors:** Xin Guan, Lujun Pan, Zeng Fan

**Affiliations:** 1International Studies College, National University of Defense Technology, Nanjing 210012, China; xguan1408@163.com; 2School of Physics, Dalian University of Technology, No. 2 Linggong Road, Ganjingzi District, Dalian 116024, China

**Keywords:** conducting polymers, electrodes, all-solid-state supercapacitors, transparency, flexibility

## Abstract

Lightweight energy storage devices with high mechanical flexibility, superior electrochemical properties and good optical transparency are highly desired for next-generation smart wearable electronics. The development of high-performance flexible and transparent electrodes for supercapacitor applications is thus attracting great attention. In this work, we successfully developed flexible, transparent and highly conductive film electrodes based on a conducting polymer, poly(3,4-ethylenedioxythiophene):poly(styrenesulfonate) (PEDOT:PSS). The PEDOT:PSS film electrodes were prepared via a simple spin-coating approach followed by a post-treatment with a salt solution. After treatment, the film electrodes achieved a high areal specific capacitance (3.92 mF/cm^2^ at 1 mA/cm^2^) and long cycling lifetime (capacitance retention >90% after 3000 cycles) with high transmittance (>60% at 550 nm). Owing to their good optoelectronic and electrochemical properties, the as-assembled all-solid-state device for which the PEDOT:PSS film electrodes were utilized as both the active electrode materials and current collectors also exhibited superior energy storage performance over other PEDOT-based flexible and transparent symmetric supercapacitors in the literature. This work provides an effective approach for producing high-performance, flexible and transparent polymer electrodes for supercapacitor applications. The as-obtained polymer film electrodes can also be highly promising for future flexible transparent portable electronics.

## 1. Introduction

As one of the most attractive power sources, supercapacitors that deliver rapid charging/discharging rates, high power densities and long lifecycles have received significant attention over the past few years [1,2,3]. High-performance supercapacitors have even been regarded as the most vital part for a wide application scope ranging, from miniaturized medical devices to large electric vehicles [4,5,6]. With the increasing desire for portable and wearable electronics, the supercapacitors in early works, which are commonly bulky, heavy and rigid, have gradually become unsatisfactory. Therefore, research interest toward developing novel energy storage devices that confer features including small size, light weight, good mechanical flexibility and even optical transparency has risen [7,8,9,10,11,12]. As the key component of a supercapacitor, the property of its electrodes is critical. To meet the aforementioned demands, the electrodes should possess not only superior electrochemical properties, but also easy processability, high mechanical flexibility and good optoelectronic performance [13].

To date, a variety of materials, including metal nanowires [14], conducting polymers [15], carbon-based nanomaterials [16,17] and nanocomposites containing transition metal oxides [18,19,20], are used to fabricate the electrodes for flexible and transparent supercapacitors. Among these materials, the hybrids of metal networks with conducting polymers have been most extensively investigated. For example, Yang et al. reported the synthesis of nickel oxide/nickel vanadate (Ni_3_V_2_O_8_) and polyaniline (PANI) composites via in situ chemical bath deposition [21]. The as-fabricated electrodes exhibited a high transparency of 71.8%, a high specific capacitance of 2565.7 F/g at 5 mV/s and a wide potential window of 1.6 V. Lee et al. developed composite electrodes consisting of poly(3,4-ethylenedioxythiophene): poly(styrenesulfonate) (PEDOT:PSS) and silver nanowires [22]. The electrodes delivered 87% transmittance and capacitance of 113 ± 18 F/g at 10 mV/s. However, the adhesion between the metal network with either the conducting polymer or the plastic substrate was weak, so the metal network was usually easily delaminated from the substrate [23]. Aiming to improve the mechanical flexibility, He et al. developed flexible and transparent film electrodes, using cellulose nanofibrils and reduced graphene oxide via a layer-by-layer self-assembly method [24]. The areal specific capacitance was 2.25 mF/cm^2^ at a current density of 0.01 mA/cm^2^. However, the carbon-based nanomaterials commonly exhibit limited optoelectronic performance [25,26,27].

Owing to the good mechanical and electrical properties, flexible and transparent electrodes made of neat conducting polymers are attracting great attention. PEDOT:PSS is considered the most successful conducting polymer because of its unique advantages, e.g., high electrical conductivity, excellent thermal stability, easy solution processability, high transparency in the visible range and possible pseudo-capacitive effects [28,29,30,31,32]. In particular, compared with the complex processing techniques for developing various inorganic-organic composite electrodes, the PEDOT:PSS film electrodes can be prepared by a simple spin-coating approach, thus allowing a high-efficient fabrication on a large scale. Due to their high electrical conductivity, PEDOT:PSS films can be applied as both the current collectors and active electrodes in flexible and transparent all-solid-state supercapacitors [33]. Lai et al. reported a flexible transparent supercapacitor based on the high-performance PEDOT:PSS films doped with ethylene glycol (EG) [33]. The highly flexible PEDOT:PSS electrodes showed high transparency of >82% and an areal specific capacitance of 4.72 mF/cm^2^ at a current density of 0.01 mA/cm^2^. The resultant all-solid-state symmetric supercapacitor yielded an areal specific capacitance of 1.18 mF/cm^2^ and an energy density of 0.105 μWh/cm^2^ at a power density of 36 μW/cm^2^. Lee et al. fabricated flexible, lightweight and coplanar transparent supercapacitor electrodes by embedding conducting silver nanofibers into PEDOT:PSS [23]. The highly flexible electrode delivered an areal capacitance of 3.64 mF/cm^2^ at 85% transmittance, while the symmetric transparent supercapacitor device exhibited an areal capacitance of 0.91 mF/cm^2^ at a scan rate of 5 mV/s and an energy density of 0.05 μWh/cm^2^ at a power density of 74.1 μW/cm^2^. However, a challenging task still remains for the development of lightweight, flexible and transparent supercapacitors with high power/energy densities.

In addition, a simple and environmental-friendly technique for the electrode fabrication is of practical significance. Previous studies have shown that the optoelectronic properties of the PEDOT:PSS films could be greatly improved via either secondary doping or a post-treatment process with solvents [34], solutions [35], ionic liquids [36], zwitterions [37], acids [29], etc. Among these approaches, the treatment using solutions consisting of common salts and solvents is particularly mild, without causing damage to the plastic substrates, e.g., polyethylene terephthalate (PET). In this regard, the PEDOT:PSS films after such a solution treatment can be promising for use as the electrodes of flexible and transparent supercapacitors.

In this work, we reported a flexible and transparent electrode with high performance based on PEDOT:PSS films. The PEDOT:PSS films were prepared through directly spin-coating the aqueous solution onto PET substrates, while their optoelectronic property was systematically investigated and modulated as a function of the spin coating cycles. The as-prepared PEDOT:PSS films were then post-treated with a solution consisting of a common inorganic salt and organic solvent. The treatment has improved both the sheet conductance and optical transmittance of the as-prepared films. An all-solid-state supercapacitor was assembled, using the treated PEDOT:PSS films as both the electrodes and current collectors. The electrochemical properties of the PEDOT:PSS electrodes and the all-solid-state device were studied in terms of the areal specific capacitance, cycling durability and mechanical flexibility. In comparison with other PEDOT-based transparent and flexible supercapacitors in literature, our device exhibited a remarkably higher power/energy density. This work offers a simple, efficient and environmental-friendly approach to fabricate high-performance electrodes for future flexible and transparent energy storage devices.

## 2. Experimental Section

### 2.1. Materials

The PEDOT:PSS aqueous solution (Clevios PH1000) was purchased from Heraeus (Shanghai, China). The concentration of PEDOT:PSS was 1.3 wt.%, and the weight ratio of PSS to PEDOT was 2.5 in solution. Isopropanol (IPA), zinc chloride (ZnCl_2_) and *N*, *N*-dimethylformamide (DMF) were purchased from Sigma-Aldrich (Shanghai, China). Phosphoric acid (H_3_PO_4_) and polyvinyl alcohol (PVA) powder were purchased from Tianjin Kaixin Chem (Tianjin, China) and Evs Chemical Technology Co. Ltd. (Dalian, China), respectively. All the chemicals were used as received. Transparent PET substrates were purchased from Dawan Plastic Electronics Company (Suzhou, China).

### 2.2. Fabrication of PEDOT:PSS Electrodes

The PEDOT:PSS electrodes was fabricated by spin-coating the PEDOT:PSS aqueous solution on PET substrates of 1.5 × 1.5 cm^2^, which were pre-treated successively by detergent, IPA and plasma cleaning (air plasma, Shenzhen Tonson Tech Automation Equipment Co., Ltd. Shenzhen, China). After spin-coating, the PEDOT:PSS films were dried at 120 °C on a hotplate for 20 min. The PEDOT:PSS electrodes with different thicknesses were prepared by repeating the spin-coating process for 1, 2, 4 and 5 cycles. The as-prepared PEDOT:PSS films were then treated with 0.1 M ZnCl_2_-DMF solution at 120 °C. After the solvent DMF was evaporated completely, the treated films were cooled down to room temperature, followed by rinsing with deionized (DI) water three times. After that, the films were dried again on the hot plate.

### 2.3. Fabrication of All-Solid-State Symmetric Supercapacitors

A gel electrolyte was prepared by mixing 1 mL H_3_PO_4_, 1g PVA powder and 9 mL DI water. The whole mixture was heated to 90 °C under stirring until the blend solution turned into a gel. The all-solid-state supercapacitors were assembled in a sandwich configuration. Typically, two PEDOT:PSS electrodes were coated on the upper and lower sides of the H_3_PO_4_/PVA gel electrolyte, respectively, and then firmly pressed together. The gel electrolyte served as both the binder and the separator. The effective area of the symmetric supercapacitor was 1.5 × 1.0 cm^2^.

### 2.4. Characterizations

The optical transmittance of the PEDOT:PSS electrodes was measured on the UV-vis-NIR spectrometer (UV-2600, Shimadzu Co., Ltd. Suzhou, China) in the visible light range (400–700 nm). The surface morphology of the PEDOT:PSS electrodes was observed on an optical microscope (PSM-1000 Motic China Group Co., Ltd. Xiamen, China). The sheet resistances of the film electrodes were measured by the van der Pauw probe technique, using a Keithley 2450 source/m. The electrical contacts were prepared on the four corners of the films, using silver paste.

### 2.5. Electrochemical Measurements

The electrochemical properties of both the PEDOT:PSS electrodes and all-solid-state devices were investigated using a CHI 760E electrochemical workstation (Chenhua Instrument Co., Shanghai, China). Cyclic voltammetry (CV), galvanostatic charge–discharge (GCD) and electrochemical impedance spectroscopy (EIS) measurements for the PEDOT:PSS electrodes were conducted in a three-electrode configuration. A 1 M sulfuric acid (H_2_SO_4_) was used as the electrolyte, while the PEDOT:PSS films, Hg/HgSO_4_ and platinum (Pt) foil served as the working, reference and counter electrodes, respectively. The EIS measurements were performed in the 10 mHz to 100 kHz frequency range with a potential amplitude of 5 mV. The electrochemical properties of the all-solid-state symmetric supercapacitor were evaluated in a two-electrode configuration by CV, GCD and EIS measurements.

The areal specific capacitances (*C*_SC_, mF/cm^2^) of the film electrodes and the symmetric supercapacitor were both calculated from their GCD curves according to the following formula:(1)$$C_{\mathrm{SC}}=\frac{I\Delta t}{A\Delta V}$$
where *I* is the discharge current, Δ*t* is the discharging time based on the scan rate, *A* is the effective area of the electrode, and Δ*V* is the potential window. The areal energy density (*E*, μWh/cm^2^) and power density (*P*, μW/cm^2^) were calculated according to the equations
(2)$$E=\frac{1}{2}C_{\mathrm{SC}}\Delta V^{2}$$

And
(3)$$P=\frac{E}{\Delta t}$$
respectively, where *C*_SC_ is the specific capacitance of the device.

## 3. Results and Discussion

Compared with other transparent supercapacitor electrodes composed of inorganic and organic blends, neat polymer film electrodes can possess various advantages, e.g., low-cost, good material uniformity and easy processability. The pristine PEDOT:PSS films directly produced from its aqueous solution usually have a very low electrical conductivity of <1 S/cm [38]. To achieve highly conductive electrodes in a mild way, the as-prepared PEDOT:PSS films were subjected to a post-treatment with 0.1 M ZnCl_2_-DMF solution. Notably, since the PEDOT:PSS films was prepared via spin-coating from its aqueous solution, and the post-treatment was performed by simple solution drop-casting, the process for producing the transparent and flexible PEDOT:PSS film electrodes is, therefore, all-solution-based, which can enable a scalable fabrication in practice. The mechanism for the electrical conductivity enhancement of PEDOT:PSS by the ZnCl_2_-DMF treatment is illustrated in Figure 1. During the ZnCl_2_-DMF treatment, the solvent DMF and the salt ions can both induce charge screening on PEDOT and PSS [39]. The charge screening effect weakens the Coulombic attraction between the PEDOT and PSS chains, thus facilitating a structural transition of the PEDOT:PSS chains from a coiled to a linear conformation. The expansion of the polymer chains consequently results in a remarkably improved electrical performance. Since the solvent and salt ions in the salt solution can play a synergetic effect on altering the polymer conformation, the salt solution treatment turns out to facilitate greater enhancement in the electrical conductivity of PEDOT:PSS than the neat solvent treatment methods, which are commonly used elsewhere [22].

For use as the flexible and transparent supercapacitor electrodes, the optoelectronic performance of the PEDOT:PSS films in terms of the sheet resistance and transmittance were measured, as shown in Figure 2. After the ZnCl_2_-DMF solution treatment, the sheet resistance of the PEDOT:PSS film prepared via only once spin-coating sharply decreases from 700 to 0.163 kΩ/sq, while its transmittance at 550 nm surprisingly increases from 89.7% to 93.9%, indicating the effectiveness of the treatment process. The sheet resistance and optical transmittance of the PEDOT:PSS film electrodes were further investigated as a function of the spin-coating cycles. As the number of the coating cycles increases from 2 to 5, the film electrode exhibits simultaneous decreases in both the sheet resistance and transparency. Typically, a PEDOT:PSS electrode prepared from 4-cycle spin-coating yields a sheet resistance of as low as 41 Ω/sq, with a transmittance of over 60% at 550 nm. These properties are quite comparable or even better than those of the PANI- and graphene-based counterparts reported in the literature [15,17].

Figure 3 displays the electrochemical properties of the PEDOT:PSS film electrodes with varying spin-coating cycles. As shown in Figure 3a, all the CV curves are nearly rectangular without redox peaks at a scan rate of 100 mV/s, indicating the typical electrical-double-layer (EDL) capacitive characteristics and good electrical conductivity of the PEDOT:PSS electrodes [40]. As the coating cycle increases from 1 to 5, the area of the closed CV loop accordingly increases. The capacitive behaviors of the PEDOT:PSS electrodes are thus improved by increasing the thickness of the PEDOT:PSS films. At the applied current density of 0.025 mA/cm^2^, all their GCD curves exhibit a nearly perfect triangular shape (Figure 3b). The small voltage drop (IR drop) suggests a decreased mass transport resistance and a good charge propagation of ions within the electrodes [41]. According to the GCD curves, the areal specific capacitances of the PEDOT:PSS electrodes prepared via spin-coating of 1, 2, 4 and 5 cycles are calculated to be 1.89, 2.27, 3.92 and 4.82 mF/cm^2^, respectively. As the discharge current density increases from 0.025 to 0.1 mA/cm^2^, their areal specific capacitances only slightly decrease to 1.52, 1.84, 3.2 and 4.02 mF/cm^2^, respectively. The corresponding capacitance retentions are 80.6%, 81.0%, 81.6%, and 83.5%, respectively, indicating the good rate capability of these electrodes (Figure 3c) [42]. The Nyquist plots as a function of the spin-coating cycles are shown in Figure 3d. At the low frequency regions, all the electrodes exhibit a steep slope, further confirming their ideal capacitive behavior [43]. At the high-frequency regions (the inset of Figure 3d), the equivalent series resistances (ESRs), which correspond to the internal resistances, are calculated to be ~3 Ω for all these electrodes.

Our previous work has shown that the post-treatment with organic solutions of inorganic salts (e.g., ZnCl_2_, NiCl_2_ and CuCl_2_) could effectively screen the Coulombic interaction between the positively charged PEDOT and negatively charged PSS. This process not only facilitates the effective PSS depletion from PEDOT:PSS, but also induces the PEDOT conformation change from a benzoid to a quinoid structure. After such a treatment, the PEDOT:PSS films could achieve a remarkable conductivity enhancement by nearly 4 orders of magnitude, from ~0.2 to ~1400 S/cm [39]. The high conductivity of the electrodes contributes to the low ESR. A low ESR has shown to be very important for reducing the IR drop of the electrodes and the assembled devices during the charging/discharging process at high current densities [13,44]. It is worth noting that the ESR of our PEDOT:PSS electrodes is significantly lower than the corresponding values of many transparent electrodes reported previously [16,45,46,47].

Given the trade-off between the optical transmittance and electrochemical storage capability, the 4-cycle spin-coated PEDOT:PSS electrode, which possesses transmittance of >60% at 550 nm, was selected as the ideal electrode for further investigation and device assembly. The optical microscope image shows that the film has a highly uniform and smooth surface morphology (Figure 4), which could be attributed to the good wettability of the PEDOT:PSS aqueous dispersion on the PET substrates. Within a potential window of −0.6–0.4 V, the CV curves of the typical 4-cycle spin-coated PEDOT:PSS electrode at scan rates of 20–500 mV/s are shown in Figure 5a. Notably, the CV curve retains a good rectangular shape, even at a high scan rate of 500 mV/s, thus suggesting the fast electron transfer and ion transfer throughout the electrode [48].

Derived from the GCD curves (Figure 5b), its specific capacitance is calculated to be 6.3 mF/cm^2^ at a current density of 0.01 mA/cm^2^, and 3.92 mF/cm^2^ at a current density of 0.1 mA/cm^2^, significantly greater than that of the electrodes prepared from the PEDOT:PSS solution doped with EG and a surfactant [33]. The cyclic stability of the 4-cycle spin-coated PEDOT:PSS electrode was assessed by GCD over 3000 cycles, as shown in Figure 5c. After 3000 charge/discharge cycles, the electrode still exhibits a capacitance retention of >90%. The corresponding GCD curves after 1000 and 3000 cycles almost overlap with the initial one (inset of Figure 5c). These results suggest the excellent cycling performance of the electrode, which is quite comparable with the reported electrochemical stability of other polymer-based counterparts [42,49,50]. The PEDOT:PSS electrode exhibits good mechanical flexibility and stable electrochemical performance at bending states. As the bending angle increases from 0 to 180°, its sheet resistance only slightly increases, i.e., by less than 20% (Figure 6). The CV and cyclic GCD measurements were also performed when the electrode was bent at a bending angle of 90°. For the electrode at normal and bending states, no significant difference could be observed in the capacitance retention and the CV and GCD curves after 500 cycles (Figure 5d), further suggesting its outstanding electrochemical stability and robust flexibility.

An all-solid-state symmetric supercapacitor was further constructed by sandwiching a H_3_PO_4_/PVA gel electrolyte between two PEDOT:PSS electrodes (Figure 7a). As shown in Figure 7b, the as-assembled supercapacitor exhibits an approximately rectangular shape in the CV curves at scan rates from 20 to 500 mV/s. The rectangular characteristic remains almost unchanged with the increasing scan rate, thereby indicating a rapid charge transport and a high rate performance of the device [42]. At current densities of 0.01–1 mA/cm^2^, all the GCD curves exhibit a symmetric triangular shape (Figure 7c). The areal specific capacitance of the supercapacitor as a function of the current density is plotted in Figure 7d. As calculated from the discharge curves, the corresponding areal specific capacitances are 1.32 and 1.14 mF/cm^2^ at the current densities of 0.01 and 1 mA/cm^2^, respectively. Such rate capability (>86% retention) is greater over those of most other polymer-based all-solid-state supercapacitors in the literature [15,23,33]. Figure 7e presents the Nyquist plot of the device. At the low-frequency regions, the curve appears to be nearly vertical, which implies an ideal capacitive behavior of the device. The absence of the semicircle at high-frequency regions indicates the good electrical contact and the rapid electron transport in the all-solid-state device [51]. The long-term cycling stability is characterized by performing 3000 GCD cycles at the applied current density of 0.2 mA/cm^2^. As shown in Figure 7f, there is no significant capacitance drop observed during the measurements. After 3000 charge/discharge cycles, the device achieves high capacitance retention of up to 96.8%. The 1st, 1000th and 3000th cycles are specifically highlighted in the insets of Figure 7f. The GCD curves after 1000 and 3000 cycles are nearly overlapped with the initial one, further demonstrating the outstanding cycling stability. To assess the flexibility of the all-solid-state supercapacitor, the CV and GCD measurements were further performed at normal and bending states. As shown in Figure 8, both the CV and GCD curves well remain in the original shapes, even at a highly bending state, revealing the excellent flexibility and mechanical robustness of the device.

The Ragone plot, which displays the relationship between energy density versus the power density, is commonly employed to evaluate the practical energy storage performance of a supercapacitor. As shown in Figure 9, the all-solid-state flexible supercapacitor based on the PEDOT:PSS electrodes delivers a high areal energy density of 0.183 μWh/cm^2^ at an areal power density of 4.98 μW/cm^2^, and maintains 0.131 μWh/cm^2^ even at 453.5 μW/cm^2^. Such power/energy densities are sufficient for powering burst communication for an integrated sensor [27]. The energy storage performance of our device is superior in comparison with those of many transparent and flexible PEDOT-based devices reported recently [23,27,33,52]. The excellent performance could be mainly attributed to the high electrical conductivity and good electrochemical property of the PEDOT:PSS films treated by the salt solution.

## 4. Conclusions

In summary, transparent, flexible and highly conductive polymer film electrodes were successfully fabricated by spin-coating the commercial PEDOT:PSS dispersion on PET substrates followed by a mild post-treatment with ZnCl_2_-DMF solution. The optical, electrical and electrochemical properties of the PEDOT:PSS film electrodes were optimized through a modulation of the cycle number of the spin-coating process. The PEDOT:PSS film electrodes obtained via 4-cycle spin-coating achieved an ideal balance between the electrochemical performance and the optical transparency, delivering a high areal specific capacitance of 3.92 mF/cm^2^ at 0.1 mA/cm^2^, a high capacitance retention of > 90% after 3000 cycles and a low ESR of only ~3 Ω with a good optical transmittance of exceeding 60%. Upon the construction of an all-solid-state transparent supercapacitor, the PEDOT:PSS film electrodes could be used as both the active electrodes and current collectors. Owing to their high electrical conductivity and good electrochemical performance, the device exhibited an areal capacitance of 1.32 mF/cm^2^, a maximum power density of 453.5 μW/cm^2^ and an energy density of 0.131 μWh/cm^2^. Such power/energy densities were remarkably larger than most of the reported flexible and transparent PEDOT-based devices so far. The approach demonstrated in this work for fabricating the PEDOT:PSS transparent electrodes is mild and easy. The as-obtained PEDOT:PSS film electrodes and devices showed excellent electrochemical performance at good optical transparency, and thus, hold great promise for use as transparent electrodes for various flexible or wearable energy storage devices.

## Figures and Tables

**Figure 1 membranes-11-00788-f001:**
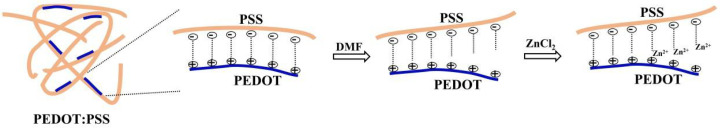
Schematic diagram of the mechanism for conductivity enhancement of PEDOT:PSS by ZnCl_2_-DMF treatment.

**Figure 2 membranes-11-00788-f002:**
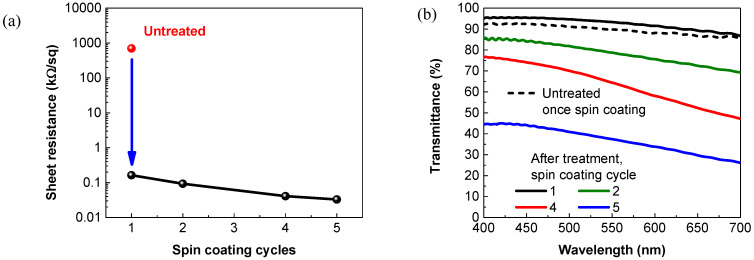
Optoelectronic property of PEDOT:PSS film electrodes. (**a**) Sheet resistances and (**b**) optical transmittance of the PEDOT:PSS film electrodes as a function of the spin-coating cycles.

**Figure 3 membranes-11-00788-f003:**
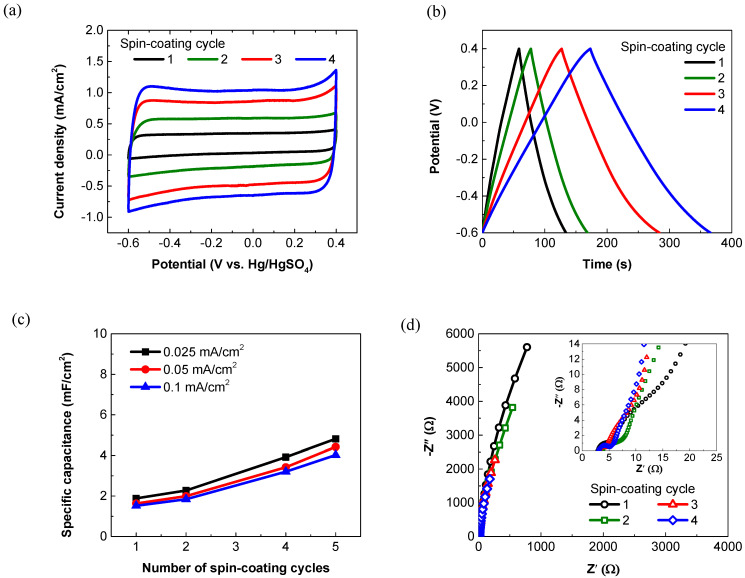
Electrochemical properties of the PEDOT:PSS electrodes as a function of the spin-coating cycles. (**a**) CV curves recorded at the scan rate of 100 mV/s. (**b**) GCD curves recorded at 0.025 mA/cm^2^. (**c**) Specific capacitance of the PEDOT:PSS electrodes spin-coated for different cycles at scan rates of 0.01–0.1 mA/cm^2^. (**d**) Nyquist plots with an enlarged scale.

**Figure 4 membranes-11-00788-f004:**
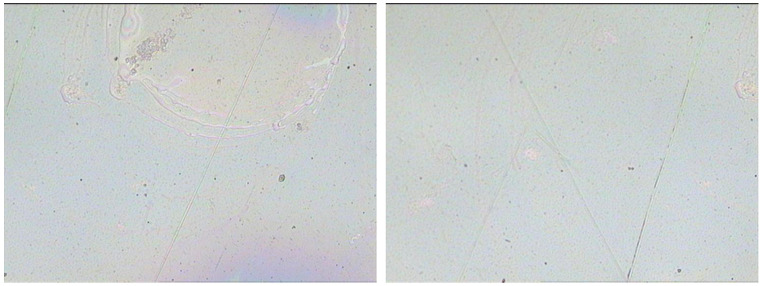
The optical microscopy images of the 4-layer PEDOT:PSS film electrode.

**Figure 5 membranes-11-00788-f005:**
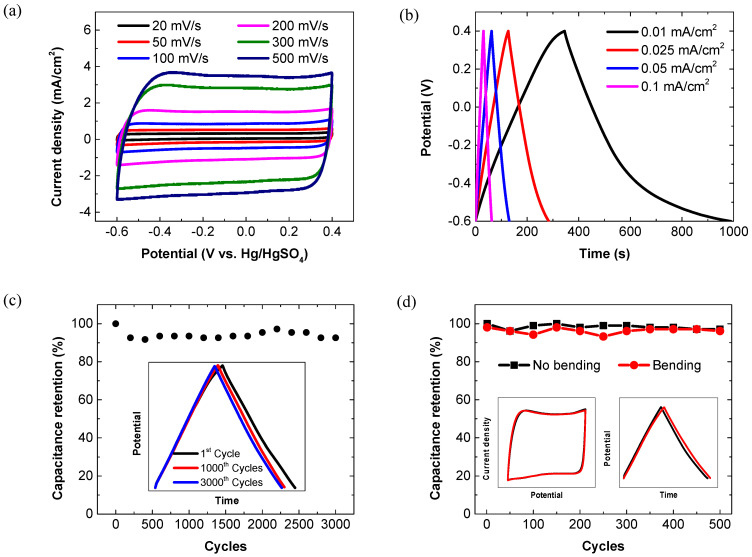
Electrochemical properties of the 4-cycle spin-coated PEDOT:PSS electrodes. (**a**) CV curves recorded at various scan rates. (**b**) GCD curves recorded at various current densities. (**c**) Cyclic stability at a current density of 0.2 mA/cm^2^. Insets show the GCD curves of the 1st, 1000th and 3000th cycles. (**d**) Cyclic stability of the electrode without bending and under bending at 90°. Insets show the comparison of the corresponding CV curves (left) and GCD curves (right) tested at normal and bending states after 500 cycles.

**Figure 6 membranes-11-00788-f006:**
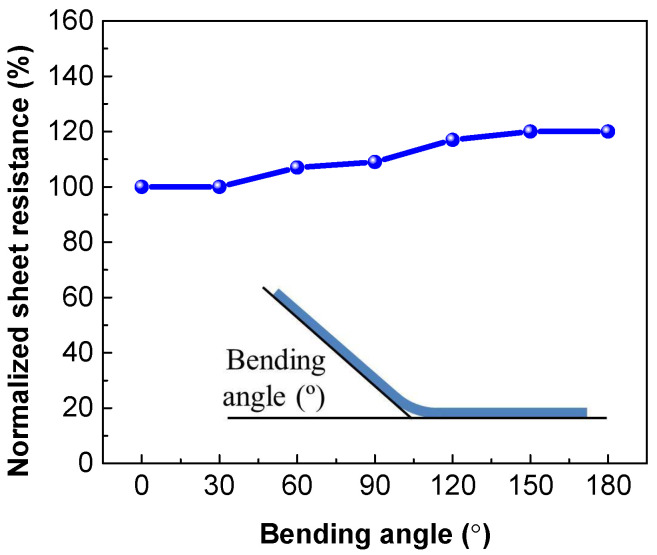
Sheet resistance of the 4-layer PEDOT:PSS film electrode as a function of the bending angle.

**Figure 7 membranes-11-00788-f007:**
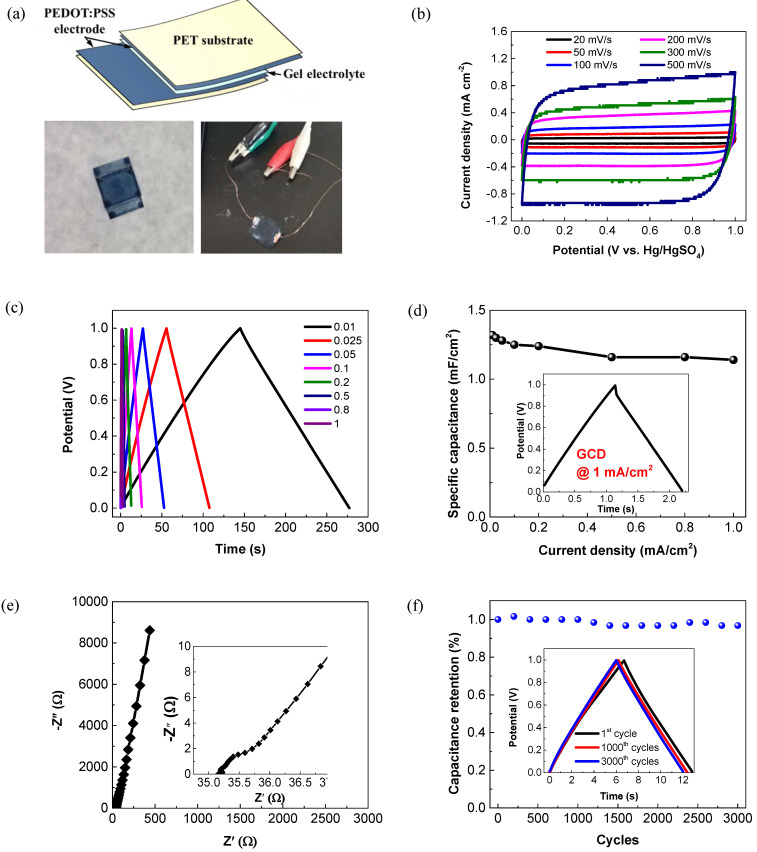
Electrochemical performance of the all-solid-state PEDOT:PSS supercapacitor. (**a**) Schematic structure and digital images of the all-solid-state PEDOT:PSS supercapacitor. (**b**) CV curves at different scan rates within a voltage window of 0–1.0 V. (**c**) GCD curves at different current densities (Unit of current density: mA/cm^2^). (**d**) Specific capacitance at different current densities. (**e**) Nyquist plot with an enlarged view in the insert. (**f**) Cycling performance at a current density of 0.2 mA/cm^2^.

**Figure 8 membranes-11-00788-f008:**
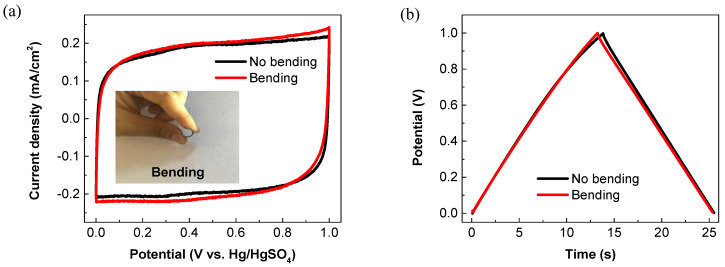
Flexibility of the all-solid-state PEDOT:PSS supercapacitor at normal and bending states. (**a**) CV curves (100 mV/s) and (**b**) GCD curves (0.1 mA/cm^2^). Inset shows the digital image of the device at bending state.

**Figure 9 membranes-11-00788-f009:**
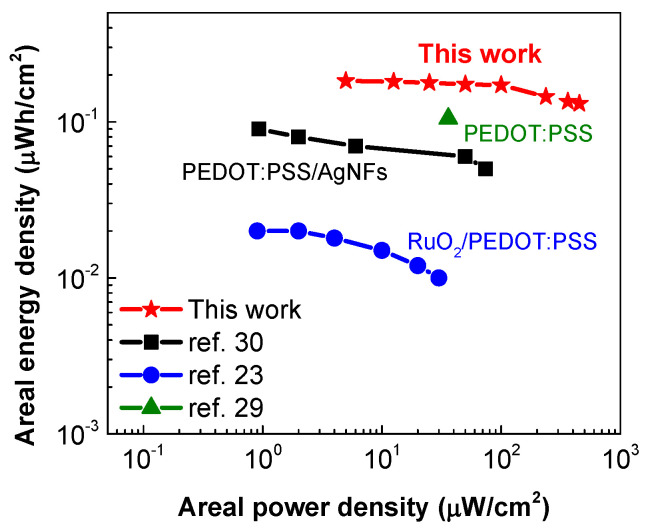
Power density versus energy density (Ragone plot) of transparent and flexible PEDOT-based symmetric supercapacitors in the literature.

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
