# Peer review of "Flexible, Transparent and Highly Conductive Polymer Film Electrodes for All-Solid-State Transparent Supercapacitor Applications"

_membranes, 2021, doi:10.3390/membranes11100788_

Round 1

Reviewer 1 Report

The authors demonstrate flexible and transparent PEDOT:PSS based all-solid-state supercapacitors. The authors successfully fabricate transparent films with high conductivity, and made solid-state supercapacitors using thoses transparent electrodes. I think this can be published after minor revision.

  1. Why PEDOT:PSS is treated by ZnCl2-DMF? The brief reason and mechanism should be mentioned.
  2. The method to calculate capacitance should be more clearly explained in the manuscript.
  3. More literatures on transparent supercapacitors should be introduced in introduction section.

Reviewer 2 Report

In the manuscript under consideration, the authors are reporting about a flexible and transparent electrode with high electrochemical performance based on a conducting polymer, poly(3,4-ethylenedioxythiophene):poly(styrenesulfonate) (PEDOT:PSS). In my opinion, is either the research has been poorly designed or has not been well explained. It sounds more of a technical report than research work.

More specific comments are found below;

The title should be revised, the authors should avoid the use of abbreviations and other unnecessary punctuations in the title.

“Abstract” part:

The abstract part needs to be revised and modified to present only the most relevant goal, idea, and get quickly to the point of the paper. It is essential to highlight the material merits and work novelty.

“Introduction” part:

While writing the introductory part, the authors focused mainly on their components. Therefore, this part should be revised with more attention to materials and other efforts used in the same field of work. Hence, it is best to expand the introduction to include:

  • Review the development of the current research problem, challenge, and current solutions
  • More attention should be paid to the other efforts and materials used in this field
  • Advantage and disadvantages leading to the current study, this means the authors highlight the development of their research and give the challenge and solution.

“The experimental section;”

All the materials used should be well explained including equipment (manufacturer’s name and location), software and statistical methods.

“Results and Discussion” part:

What is the novelty of materials used?

The conclusion part should be revised on the basis of the material, the objective, the main results obtained, as well as clarification of the future applicability

In the manuscript under consideration, the authors are reporting about a flexible and transparent electrode with high electrochemical performance based on a conducting polymer, poly(3,4-ethylenedioxythiophene):poly(styrenesulfonate) (PEDOT:PSS). In my opinion, is either the research has been poorly designed or has not been well explained. It sounds more of a technical report than research work.

More specific comments are found below;

The title should be revised, the authors should avoid the use of abbreviations and other unnecessary punctuations in the title.

“Abstract” part:

The abstract part needs to be revised and modified to present only the most relevant goal, idea, and get quickly to the point of the paper. It is essential to highlight the material merits and work novelty.

“Introduction” part:

While writing the introductory part, the authors focused mainly on their components. Therefore, this part should be revised with more attention to materials and other efforts used in the same field of work. Hence, it is best to expand the introduction to include:

  • Review the development of the current research problem, challenge, and current solutions
  • More attention should be paid to the other efforts and materials used in this field
  • Advantage and disadvantages leading to the current study, this means the authors highlight the development of their research and give the challenge and solution.

“The experimental section;”

All the materials used should be well explained including equipment (manufacturer’s name and location), software and statistical methods.

 “Results and Discussion” part:

What is the novelty of materials used?

The conclusion part should be revised on the basis of the material, the objective, the main results obtained, as well as clarification of the future applicability

In the manuscript under consideration, the authors are reporting about a flexible and transparent electrode with high electrochemical performance based on a conducting polymer, poly(3,4-ethylenedioxythiophene):poly(styrenesulfonate) (PEDOT:PSS). In my opinion, is either the research has been poorly designed or has not been well explained. It sounds more of a technical report than research work.

More specific comments are found below;

The title should be revised, the authors should avoid the use of abbreviations and other unnecessary punctuations in the title.

“Abstract” part:

The abstract part needs to be revised and modified to present only the most relevant goal, idea, and get quickly to the point of the paper. It is essential to highlight the material merits and work novelty.

“Introduction” part:

While writing the introductory part, the authors focused mainly on their components. Therefore, this part should be revised with more attention to materials and other efforts used in the same field of work. Hence, it is best to expand the introduction to include:

  • Review the development of the current research problem, challenge, and current solutions
  • More attention should be paid to the other efforts and materials used in this field
  • Advantage and disadvantages leading to the current study, this means the authors highlight the development of their research and give the challenge and solution.

“The experimental section;”

All the materials used should be well explained including equipment (manufacturer’s name and location), software and statistical methods.

 “Results and Discussion” part:

What is the novelty of materials used?

The conclusion part should be revised on the basis of the material, the objective, the main results obtained, as well as clarification of the future applicability

In the manuscript under consideration, the authors are reporting about a flexible and transparent electrode with high electrochemical performance based on a conducting polymer, poly(3,4-ethylenedioxythiophene):poly(styrenesulfonate) (PEDOT:PSS). In my opinion, is either the research has been poorly designed or has not been well explained. It sounds more of a technical report than research work.

More specific comments are found below;

The title should be revised, the authors should avoid the use of abbreviations and other unnecessary punctuations in the title.

“Abstract” part:

The abstract part needs to be revised and modified to present only the most relevant goal, idea, and get quickly to the point of the paper. It is essential to highlight the material merits and work novelty.

“Introduction” part:

While writing the introductory part, the authors focused mainly on their components. Therefore, this part should be revised with more attention to materials and other efforts used in the same field of work. Hence, it is best to expand the introduction to include:

  • Review the development of the current research problem, challenge, and current solutions
  • More attention should be paid to the other efforts and materials used in this field
  • Advantage and disadvantages leading to the current study, this means the authors highlight the development of their research and give the challenge and solution.

“The experimental section;”

All the materials used should be well explained including equipment (manufacturer’s name and location), software and statistical methods.

 “Results and Discussion” part:

What is the novelty of materials used?

The conclusion part should be revised on the basis of the material, the objective, the main results obtained, as well as clarification of the future applicability

In the manuscript under consideration, the authors are reporting about a flexible and transparent electrode with high electrochemical performance based on a conducting polymer, poly(3,4-ethylenedioxythiophene):poly(styrenesulfonate) (PEDOT:PSS). In my opinion, is either the research has been poorly designed or has not been well explained. It sounds more of a technical report than research work.

More specific comments are found below;

The title should be revised, the authors should avoid the use of abbreviations and other unnecessary punctuations in the title.

“Abstract” part:

The abstract part needs to be revised and modified to present only the most relevant goal, idea, and get quickly to the point of the paper. It is essential to highlight the material merits and work novelty.

“Introduction” part:

While writing the introductory part, the authors focused mainly on their components. Therefore, this part should be revised with more attention to materials and other efforts used in the same field of work. Hence, it is best to expand the introduction to include:

  • Review the development of the current research problem, challenge, and current solutions
  • More attention should be paid to the other efforts and materials used in this field
  • Advantage and disadvantages leading to the current study, this means the authors highlight the development of their research and give the challenge and solution.

“The experimental section;”

All the materials used should be well explained including equipment (manufacturer’s name and location), software and statistical methods.

 “Results and Discussion” part:

What is the novelty of materials used?

T
